# Outcome One Year after Acetabular Rim Extension Using a Customized Titanium Implant for Treating Hip Dysplasia in Dogs

**DOI:** 10.3390/ani14162385

**Published:** 2024-08-17

**Authors:** Irin Kwananocha, Joëll Magré, Amir Kamali, Femke Verseijden, Koen Willemsen, Yuntao Ji, Bart C. H. van der Wal, Ralph J. B. Sakkers, Marianna A. Tryfonidou, Björn P. Meij

**Affiliations:** 1Department of Clinical Sciences, Faculty of Veterinary Medicine, Utrecht University, Yalelaan 1, 3584 CL Utrecht, The Netherlands; i.kwananocha@uu.nl (I.K.); s.a.kamali@uu.nl (A.K.); f.verseijden@uu.nl (F.V.); m.a.tryfonidou@uu.nl (M.A.T.); 2Research and Academic Service, Faculty of Veterinary Medicine, Kasetsart University, 50 Ngamwongwan Rd., Lat Yao, Chatuchak, Bangkok 10900, Thailand; 3Department of Orthopedics, University Medical Center Utrecht, Heidelberglaan 100, 3584 CX Utrecht, The Netherlands; j.magre-2@umcutrecht.nl (J.M.); k.willemsen-3@umcutrecht.nl (K.W.); b.c.h.vanderWal@umcutrecht.nl (B.C.H.v.d.W.); r.sakkers@umcutrecht.nl (R.J.B.S.); 43D Lab UMC Utrecht, Division of Surgical Specialties, University Medical Center Utrecht, Heidelberglaan 100, 3584 CX Utrecht, The Netherlands; 5Department of Earth Sciences, Faculty of Geosciences, Utrecht University, Princetonlaan 8a, 3584 CB Utrecht, The Netherlands; y.ji@uu.nl

**Keywords:** hip dysplasia, dog, customized implant, 3D printed implant, femoral head coverage, acetabular rim extension, ACE-X

## Abstract

**Simple Summary:**

Hip dysplasia is prevalent among young dogs, affecting approximately 16% of the overall dog population, causing pain and exercise intolerance, and potentially leading to severe osteoarthritis. Current surgical interventions, such as pelvic osteotomies and total hip replacement, have limitations and a high level of complications. To address this, a personalized 3D-printed titanium implant was developed for dorsal acetabular rim extension (ACE-X), enhancing hip joint coverage and reducing hip laxity. In a previous short-term study spanning 3 months, the ACE-X implant demonstrated promising results: improved femoral head coverage, decreased pain-related activities based on owner questionnaires, and facilitated rapid recovery. This study aimed to offer a thorough analysis of clinical outcomes after a one-year follow-up period. The analysis encompassed radiographic measurements, force plate analysis, owner questionnaires, and an assessment of bone–implant osseointegration, as well as the identification and documentation of any complications encountered during the study period. By evaluating the long-term efficacy and safety of the ACE-X implant, this research contributes valuable insights into the management of hip dysplasia in dogs, potentially offering a novel and effective treatment option for this prevalent orthopedic condition.

**Abstract:**

The acetabular rim extension (ACE-X) implant is a custom-made three-dimensionally printed titanium device designed for the treatment of canine hip dysplasia. In this study, 34 dogs (61 hips) underwent ACE-X implantation, and assessments were conducted using computed tomography, force plate analysis, Ortolani’s test, and the Helsinki chronic pain index (HCPI) questionnaires at five intervals: the pre-operative day, the surgery day, and the 1.5-month, 3-month, and 12-month follow-ups. Statistically significant increases in femoral head coverage with a negative Ortolani subluxation test were observed immediately after surgery and persisted throughout the study. Osteoarthritis (OA) scores remained stable, but osteophyte size significantly increased between the surgery day and the 12-month follow-up, especially in hips with a baseline OA score of 2 compared to those with a score of 1. The force plate data showed no significant changes during the study. The HCPI demonstrated a significant decrease in pain score from pre-operative value to six-week follow-up and gradually decreased over time. Major complications were identified in six hips (9.8%) of four dogs. In conclusion, the ACE-X implant effectively increased femoral head coverage, eliminated subluxation, and provided long-term pain relief with minimal complications, benefiting over 90% of the study population. The study supports the ACE-X implant as a valuable alternative treatment for canine hip dysplasia.

## 1. Introduction

Canine hip dysplasia (CHD) is a prevalent developmental orthopedic condition that affects young dogs, especially those of medium to large breeds, with an overall prevalence of 16% [1]. CHD leads to instability (laxity) of the hip joint and hip pain. Owners may observe abnormal gait and lameness after exercise, typically becoming noticeable between 4 and 10 months of age [2]. This condition occurs due to continuous abnormal movement of the femoral head (ball), which causes deformation of the acetabulum (socket). The long-term consequence of this joint laxity is the gradual loss of cartilage and the early onset of osteoarthritic changes in the joint. Conservative management alone cannot halt the progression of clinical dysplastic disease; therefore, early surgical intervention is necessary to restore the joint’s normal weight-bearing forces and prevent or slow down osteoarthritic changes.

Current surgical options for early-stage CHD include juvenile pubic symphysiodesis (JPS), double or triple pelvic osteotomy (DPO/TPO), intertrochanteric osteotomy, and dorsal acetabular rim arthroplasty (DARthroplasty). These techniques aim to reduce hip joint laxity by increasing the weight-bearing surface, thereby preventing further joint damage [3,4,5,6,7]. However, each of these techniques has its inherent disadvantages, such as upper age limits for surgery (JPS and DPO), invasiveness (DPO/TPO), a risk of complications, and suboptimal outcomes in all procedures [3,8,9,10,11].

To offer an alternative treatment option with immediate reduction of hip laxity, faster recovery time, and a lower risk of harm, a novel personalized surgical technique for early-stage CHD was developed. This new technique utilizes a patient-specific three-dimensional (3D) printed shelf implant for dorsal acetabular rim extension (ACE-X). The ACE-X implant is designed based on computed tomography (CT) imaging of the pelvis and hip joints of individual dogs and is used to extend the dorsal acetabular rim optimally and provide sufficient coverage of the femoral head. In previous proof-of-concept pilot studies [12] involving asymptomatic dogs with radiographic hip dysplasia (*n* = 3), and in a short-term (3-month) clinical trial [13] in these 34 dogs with hip dysplasia (HD), the ACE-X implant reduced hip laxity and restored the coverage of dysplastic hip joints. This new technique without the use of osteotomies was associated with low postoperative morbidity and fast gait recovery, allowing simultaneous bilateral procedures in one operative session. The present study was designed to investigate the outcomes of the ACE-X implant on hip laxity and osteoarthritic changes in client-owned dogs with HD over an extended duration. Clinical examinations, gait analysis using force plates, owners’ questionnaires, and CT scans were used to evaluate the efficacy of the ACE-X implant up to 12 months postoperatively. Furthermore, successful osseointegration of the ACE-X implant, which is crucial for implant stability and long-term implant survival, was investigated.

## 2. Materials and Methods

### 2.1. Study Design and Overview

This study is a prospective non-randomized unblinded self-controlled clinical trial involving client-owned dogs that underwent surgery for ACE-X implantation at the Department of Clinical Sciences, Faculty of Veterinary Medicine, Utrecht University between 2019 and 2022. Approval for the study was obtained from the Veterinary Clinical Studies Committee, Utrecht University, Utrecht, The Netherlands. Throughout the study, the dogs remained under the supervision and care of their owners, who were informed about the study’s objectives, the surgical ACE-X procedure, available alternative treatments, the treatment plan, and all potential complications associated with ACE-X treatment (e.g., infection, implant failure, neurological deficits, and others). All owners provided informed written consent before their dogs participated in the study.

A total of 34 dogs (61 studied hips) were monitored during a 12-month follow-up period. During this period, the dogs were examined at 5 designated time points: the pre-operative day (visit 1), the day of surgery (visit 2), 1.5-month follow-up (visit 3), 3-month follow-up (visit 4), and 12-month follow-up (visit 5). Different clinical observations, including imaging, force plate gait analysis, Ortolani’s test, and the Helsinki chronic pain index (HCPI) questionnaire, were recorded at each visit (Table 1).

Client-owned dogs that were older than 6 months and that exhibited a positive Ortolani’s test result under sedation and radiographic signs of hip dysplasia within the Fédération Cynologique Internationale (FCI) score range of B (borderline) to E (severe) [14,15,16,17], as well as radiographic osteoarthritis (OA) scores falling between 0 (osteophyte size 0 mm) and 1 (osteophyte size 0.1–2 mm) [18], were considered eligible for participation in the study. The FCI and intake OA scores were categorized based on preoperative radiographs obtained during the inclusion process, either conducted at the referring hospital or at Utrecht University. Individuals were excluded from study participation if they exhibited an open acetabular growth plate, a negative Ortolani’s test result, moderate to severe osteoarthritis (indicated by an intake OA score of ≥2 or osteophyte size of >2.0 mm), systemic disease, luxoid hip condition, neurological deficits, or a history of previous hip surgery. Preoperative CT planning was performed for the development of a personalized ACE-X implant. Mimics (version 24, Materialise, Leuven, Belgium) was used for segmentation of the DICOM files, and 3-Matic (version 16, Materialise, Leuven, Belgium) was used for implant design at the 3D Lab of University Medical Centre Utrecht, as described previously by Willemsen et al. and Kwananocha et al. [12,13,19]. The ACE-X implant was positioned extracapsular of the hip joint, over the upper edge of the dorsal acetabular rim, and secured using four screws on the iliac body (Figure 1). The duration between the day the preoperative CT was performed (visit 1) and the day of surgery (visit 2) was set as the lead time.

### 2.2. Radiographic and Clinical Assessment

#### 2.2.1. Radiographic Measurements

The Norberg angle (NA), the linear percentage of femoral head overlap (LFO), and the percentage of femoral head coverage (PC) were evaluated from coronal CT scans, following the methodology outlined in an earlier short-term report [13] during visits 1, 2, 4, and 5. Concurrently, the progression of OA was assessed based on the maximal thickness of osteophytes detected at three distinct locations on the CT: the cranial and caudal acetabular rim, and the femoral neck. These assessments were conducted using both coronal and transverse CT scans. The OA scoring system consisted of 4 distinct categories: OA score 0 (osteophyte size of 0 mm), 1 (osteophyte size 0.1 to 2 mm), 2 (osteophyte size 2.1 to 5 mm), and 3 (osteophyte size > 5 mm) [18]. The baseline value for the OA score and the measured osteophyte size were established from the immediate postoperative CT scan at visit 2. The radiographic measurements were performed using the Xero Viewer software (version 8.2.2.050, AGFA HealthCare, Mortsel, Belgium) by a single observer (I.K.).

#### 2.2.2. Force Plate Gait Analysis

Ground reaction forces (GRFs) were assessed using a single force plate (Kistler type 9261, Kistler Instrument AG, Winterthur, Switzerland), embedded at the midpoint of an 11-meter-long walkway during visits 1, 3, 4, and 5. The dogs were led on a leash across the force plate, maintaining a consistent velocity between 0.8 and 1.3 m/s. For each limb’s side, the first 4 to 10 valid trials were recorded and then averaged. A valid trial was defined as the forelimb being succeeded by the ipsilateral hind limb contacting the force plate during the dog’s walking movement. The GRFs, including breaking force, propulsion force, peak vertical force (PVF), vertical impulse (VI), and pelvic (P) and thoracic (T) indices (P/T index), were calculated and subsequently normalized relative to the dog’s body weight.

#### 2.2.3. The Helsinki Chronic Pain Index (HCPI) Questionnaire

During visits 1, 3, 4, and 5, owners were provided with the HCPI questionnaire [20]. This survey consisted of 11 inquiries, each utilizing a 5-point descriptive scale (scoring from 0 to 4). Subsequently, an overall score was calculated, ranging from 0 to 44. Scores falling between 0 and 1 for each question indicated typical dog behavior or movement, while scores ranging from 2 to 4 suggested atypical behavior or movement. A total score of less than 6 denoted the absence of pain, while a cumulative score exceeding 11 signified the presence of chronic pain. Scores between 6 and 11 constituted an indeterminate zone, as established by previous studies [20,21]. In this study, each dog’s overall index score was transformed into a percentage by dividing it by the maximum possible score, based on the answered questions. This transformation allowed for further analysis, wherein a sum score over 25% indicated chronic pain, a total score under 13.6% indicated the absence of pain, and scores between 13.6% and 25% were categorized as inconclusive.

#### 2.2.4. Ortolani’s Test

Ortolani’s test was performed while the dog was in lateral recumbency, under either sedation or general anesthesia, as required for the CT scans, radiographs, and surgery during each of the five scheduled visits.

### 2.3. Histopathology and Bone Implant Contact

In cases where dogs experienced complications necessitating revision surgery, the ACE-X implants were removed and the soft tissues surrounding the hip joint were collected for histopathological examination, as described below. When a total hip replacement (THR) was performed, the femoral head was obtained for histopathological analysis after femoral head ostectomy. Soft-tissue biopsies were fixed in 10% neutral buffered formalin (NBF, pH 7.26) for 48 h, then processed and embedded in paraffin. Formalin-fixed femoral head specimens underwent decalcification in 10% ethylenediaminetetraacetic acid (EDTA) for 3 months. The paraffin-embedded soft tissue and bone samples were sectioned into 5-µm-thick slices and stained with hematoxylin and eosin (H&E). Histological sections were examined by an independent pathologist (A.K.), utilizing light microscopy (Olympus BX51; Olympus, Tokyo, Japan). To assess osseointegration, the removed ACE-X implants were preserved in 10% neutral buffered formalin for a minimum of 6 months. Subsequently, these specimens underwent analysis using a three-dimensional (3D) X-ray microscope (Zeiss Xradia 610 Versa, Zeiss, Oberkochen, Germany), as well as histopathological examination.

For imaging, the implants were affixed to a turntable that was capable of vertical rotation, enabling optimal sample positioning. CT imaging was conducted using a Zeiss Xradia 610 Versa 3D X-ray microscope (Zeiss, Oberkochen, Germany) situated in the Multi-scale Imaging and Tomography Facility (MINT) at Utrecht University. Multiple sets of projections were acquired for each sample, with the number of projections per scan ranging from 1201 to 2401, to achieve a high signal-to-noise ratio within an acceptable scanning duration. Image reconstructions were performed using the ‘Reconstructor Scout-and-Scan’ software (version 16.0, Zeiss, Oberkochen, Germany), with pixel sizes ranging from 2.4 µm to 7.74 µm. Incident X-rays, generated using a voltage of 140 kV and 21 W power without a source filter, optimized the transmittance of the projection images, maximizing the grey-level contrast between the metal implant, bone tissue, and air, making them clearly distinguishable in the reconstructed images. Image thresholding and figure plotting were conducted using the TXM3DViewer software (version 1.2.10, Zeiss, Oberkochen, Germany).

For histopathology, the formalin-fixed explanted implant and the interconnected bone were dehydrated using a graded series of ethanol solutions. Subsequently, the sample was infiltrated with a plastic embedding substance (polymethyl methacrylate (PMMA) monomers and initiators) in a desiccator environment to achieve proper dehydration and optimum penetration. The embedding process involved carefully placing the dehydrated and PMMA-infiltrated specimen into the embedding mold, followed by polymerization through temperature and cure-time control. A 5-mm section was removed from one side of the block using a diamond-coated saw to access the middle of the sample and verify the porous structure of the explanted implant. The specimen was then sliced to 80 µm in thickness using a saw microtome (Leica SP 1600, Leica Microsystems, Wetzlar, Germany). The sections were stained with methylene blue and acid fuchsin for evaluation under light microscopy (Olympus BX53, Olympus, Tokyo, Japan).

### 2.4. Statistical Analysis

Normality was assessed using a Q-Q plot (SPSS version 28, IBM, NY, USA). Radiographic measurements (i.e., NA, LFO, PC, and osteophyte size), HCPI, and GRFs were analyzed using generalized linear mixed models. The results of the OA scores were evaluated using Friedman’s related-sample two-way analysis of variance. The Kruskal–Wallis test was employed to assess changes in osteophyte size over time among dogs with different baseline OA scores and FCI scores. Bonferroni post hoc tests were conducted to correct for differences between time periods and OA score groups. Values of *p* < 0.05 were considered statistically significant.

## 3. Results

The patient demographics have previously been detailed in an earlier publication about the short-term results [13]. In summary, this study involved 34 client-owned dogs, encompassing a total of 61 dysplastic hips. The cohort comprised 24 males and 10 females, with a median age of 12 months (7–38 months) and a median body weight of 27.3 kg (12–86 kg). Among them, 7 dogs underwent unilateral hip surgery, while 27 dogs underwent surgery on bilateral hips. Among the dogs with bilateral hip surgery, seven underwent the procedure in two separate sessions, with a median interval of 92 days (range: 56–524 days) between the two sides.

The number of various examinations conducted at different time points, including the preoperative day (visit 1), on the day of surgery (visit 2), and the 1.5-month (visit 3), 3-month (visit 4), and 12-month (visit 5) follow-ups (Table 2, Figure 2). The median lead time between the preoperative day and the day of surgery was 74 days (46–158 days). The median durations between the day of surgery and the 1.5-month, 3-month, and 12-month follow-ups were 45 days (range: 24–92 days), 99 days (range: 68–289 days), and 386 days (range: 270–553 days), respectively. Over the 12-month study period, five hips (in three dogs) were excluded at 6 months (hip no. 34), 7 months (hip no. 37), 9 months (hip no. 23), 10 months (hip no. 25), and 12 months (hip no. 24) after surgery, due to implant failure (hip nos. 23, 34, and 37) and severe osteoarthritis progression (hip no. 24 and 25). The other hips showed positive outcomes over the 12-month duration, with a representative example demonstrated in Figure 2.

### 3.1. Assessment of Hip Joint Laxity Testing (Ortolani’s Test)

At intake, all 61 hips exhibited a positive Ortolani’s test result; however, 6 hips returned a negative Ortolani’s test result during preoperative evaluation on the day of surgery. After ACE-X implantation, two hips exhibited a positive Ortolani’s test result immediately postoperative (hip no. 36) and at the 1.5-month follow-up (hip no. 56). By the 12-month follow-up, hip number 56 had transitioned to a negative Ortolani’s test result. However, another hip (hip no. 53) displayed a positive Ortolani’s test result at the 12-month follow-up (visit 5). Thus, the presence of a positive Ortolani’s test result was noted in 2 out of the 55 hips during visit 5.

### 3.2. Radiographic Outcomes

The FCI score was determined by two national hip dysplasia screening panelists (M.T. and B.M.) using ventrodorsal hip extension radiographs that were taken preoperatively, revealing FCI scores of B (7 hips), C (8 hips), D (22 hips), and E (24 hips). The NA, LFO, and PC measurements using the coronal CT significantly increased at visits 2, 4, and 5 compared to visit 1 (*p* < 0.001) (Table 3).

The median OA scores and mean (±SD) osteophyte size at visit 1 (1; range 0–2 and 1.26 ± 0.84 mm, respectively) were significantly increased when compared to visit 2 (1; range 0–2 and 1.86 ± 1.28 mm, respectively) (*p* < 0.001 and *p* < 0.05), as previously reported in the short-term results [13]. This study assessed the OA scores based on immediate post-operative CT scans at visit 2, which served as the baseline value. The median OA score did not significantly change between visit 2 (1; range 0–2), visit 4 (2; range 0–3), and visit 5 (2; range 0–3) (*p* = 0.115) (Figure 3A). However, the mean (±SD) osteophyte size increased significantly between visit 2 (1.8 ± 1.3 mm) and visit 5 (2.7 ± 2.0 mm) (*p* = 0.028) (Figure 3B). The average (±SD) increases in osteophyte size over the 12-month period from visit 2 to visit 5 significantly varied between the group with a baseline OA score of 2 (1.6 ± 1.3 mm) and the group with a baseline OA score of 1 (0.2 ± 0.6 mm) (*p* < 0.001). However, there was no significant difference observed between the group with a baseline OA score of 1 (0.2 ± 0.6 mm) and the group with a baseline OA score of 0 (0.8 ± 1.2 mm) (*p* = 0.181), as illustrated in Figure 3C. Additionally, there was no significant difference observed in the mean increase in osteophyte size between hips with FCI scores of B (0.8 ± 1.4 mm), C (0.5 ± 0.9 mm), D (0.6 ± 0.9 mm), and E (1.6 ± 1.4 mm) (*p* = 0.097), as depicted in Figure 3D.

### 3.3. Kinetic Gait Analysis

The GRFs were measured preoperatively (*n* = 51) and at the 1.5-month (*n* = 54), 3-month (*n* = 46), and 12-month (*n* = 44) follow-ups. A significant change was not observed between each visit for PVF (*p* = 0.166), the P/T index (*p* = 0.067), VI (*p* = 0.705), and breaking force (*p* = 0.726). However, propulsion force differed significantly between the preoperative evaluation (visit 1) and the follow-up evaluations at 3 months (visit 4) and 12 months (visit 5) (*p* = 0.007) (Table 3).

### 3.4. Owner Assessment of Pain-Related Behavior

Owner assessments of pain-related behavior were collected through HCPI questionnaires at various time points for each individual dog, regardless of whether they had undergone unilateral or bilateral ACE-X implantation. The distribution of returned questionnaires is presented in Table 2. Pain-related behavior significantly reduced over time, when comparing the preoperative assessment (30%) with the assessments at 1.5 months (22%), 3 months (19%), and 12 months (18%) of follow-up (*p* = 0.002) (Table 3).

### 3.5. Complications

In the previous short-term study, three hips (4.9%) had perioperative complications, including problems with imperfect implant positioning in two hips and screw misalignment in one hip [13], whereas nine hips (14.8%) had postoperative issues in this long-term study. Regarding postoperative complications, three hips (hip nos. 42, 46, and 47) from two dogs, accounting for 4.9% of the cases, experienced minor issues, notably intermittent lameness that could be attributed to OA progression. These cases responded well to pain medication and did not require revision surgery during the study period. Major complications were observed in six hips (9.8%) from four dogs (Table 4), including ACE-X implant failure (hip no. 34), screw breakage (hip nos. 23 and 37), substantial osteoarthritis progression with severe lameness (hip nos. 24 and 25), and osteomyelitis due to septic arthritis (hip no. 53). These major complications occurred at various intervals post-operation, ranging from 6 to 17 months following the initial surgery.

In one case, implant failure occurred bilaterally (hip nos. 23 and 34) in a dog that underwent staged bilateral ACE-X implantation, with a two-month interval between each hip procedure (Figure 4A). Unfortunately, the dog experienced bilateral humeral condyle fractures five months post-operation on the second hip (hip no. 34), requiring plate and screw fixation (Figure 4B). Subsequently, the ACE-X implant in the right hip (hip no. 34) and screws in the left hip (hip no. 23) broke and severe OA progression was evident in both hips (Figure 4C), necessitating revision surgery involving implant removal and total hip replacement (THR) at 9 months and 15 months post ACE-X implantation (Figure 4D), respectively (Table 4: Dog 1).

Another dog was initially scheduled for bilateral ACE-X surgery in a single session but experienced a dislocated hip on the day of surgery (visit 2). Consequently, unilateral ACE-X implantation (hip no. 37) was performed, and the contralateral limb underwent THR one month later. Approximately seven months post-ACE-X implantation, the dog developed lameness in the ACE-X-treated limb due to four fractured screws, necessitating revision surgery involving implant removal and THR (Table 4: Dog 2).

A third dog underwent single-stage bilateral hip surgery (Figure 5B) but experienced significant deterioration in the left hip (hip no.25) approximately 10 months post-ACE-X surgery (Figure 5C), resulting in severe osteoarthritis and pronounced lameness. Subsequent implant removal and THR were performed. Two months later, similar signs emerged in the contralateral hip (hip no. 24), leading to revision surgery at 22 months post-initial ACE-X implantation (Figure 5D and Table 4: Dog 3).

The last dog initially responded well to bilateral ACE-X implantation for seven months but subsequently developed intermittent lameness in the right hind leg, which was attributed to a chronic urinary tract infection. Over the course of 7 to 17 months post-ACE-X surgery, the dog underwent surgical cystotomy twice for calculi removal and experienced recurring chronic cystitis episodes coinciding with lameness. The administration of antibiotics significantly improved the lameness signs; however, at 17 months post-surgery, consistent lameness persisted in the right hind leg. A CT scan revealed the characteristic signs of infectious arthritis in the right femoral head and acetabulum (hip no. 53), prompting arthrocentesis and bacterial culture tests. *Staphylococcus pseudointermedius* was isolated from the joint fluid and urine samples, leading to a recommendation of staged revision surgery involving implant removal and joint lavage, followed by THR (Table 4: Dog 4).

### 3.6. Histopathology and 3D X-ray Microscopy

Histopathological examination and 3D X-ray microscopy were utilized to analyze the femoral heads, synovial tissues, and ACE-X implants obtained from the revision hip surgeries. The examination revealed consistent findings across the specimens. The femoral heads showed extensive cartilage loss, particularly in the region where the acetabular rim extension of the implant was located. Additionally, the necrosis of denuded subchondral bone with empty lacunae, as well as the degeneration of adjacent hyaline cartilage (characterized by fibrillation and fissures), were evident (Figure 6). Histopathological analysis of the synovial tissues demonstrated various alterations, including edema, hyperplasia of the synovial lining cells, variable degrees of stromal activation, and the formation of new blood vessels (Figure 7).

Upon gross examination, bone integration was observed in the porous surface of the bone attachment part of the removed ACE-X implants at 9 months (hip no. 34) and 22 months (hip no. 24) post-surgery, whereas this integration was not observed in the specimen explanted at 7 months (hip no. 37). Three-dimensional X-ray microscopy unveiled bone density within the porous layer of the implant extracted from hip no. 34 at 9 months and hip no. 24 at 22 months post-surgery, while no such density was detected in the control implant (Figure 8B). The original gray values were converted into 3D visualizations with color representation using the TXM3DViewer software (version 1.2.10, Zeiss, Oberkochen, Germany). Each color represented a different attenuation level of the specimen, with red indicating the highest attenuation, followed by orange, yellow, green, and blue in descending order of linear attenuation coefficient. In the control implant, red denoted the highest-density area of the implant, while yellow and green represented lower-density areas. In the removed implants, similar to previous studies, light and dark blues were observed in the porous surface, indicating the density of bone tissue (Figure 8C) [22]. Additionally, histological sections of the removed implant from hip no. 24 showed the presence of newly formed bone, characterized by a spongy architecture consisting of trabeculae with interstitial bone marrow spaces (Figure 9).

## 4. Discussion

Hip dysplasia significantly reduces the quality of life for affected dogs due to the inherent laxity of the dysplastic hip, leading to lameness, exercise intolerance, and limb impairment. This study presents the one-year outcomes of a recently introduced surgical treatment for hip dysplasia using patient-specific 3D-printed titanium implants to extend the dorsal acetabular rim [13]. The ACE-X implant demonstrated the effective retention of functionality by expanding the femoral head coverage, correcting hip laxity, and reducing the pain-related exercise limitations based on owner questionnaires during long-term monitoring.

At the 12-month follow-up, joint laxity, as evidenced by a positive Ortolani subluxation test, was only observed in one hip (hip no. 53), which had consistently shown negative results post-surgery but developed infectious arthritis secondary to a chronic bladder infection. The change to a positive Ortolani’s test result is most likely linked to septic arthritis, leading to increased joint fluid and hip joint laxity [23]. Conversely, another hip (hip no. 56) that initially exhibited a positive Ortolani’s test result at 1.5 months post-surgery displayed a negative result at the 12-month follow-up. This transformation could be attributed to joint capsule thickening [23]. However, with only two exceptions (hip nos. 53 and 36), all other dysplastic hips (96.7%) were stable immediately after ACE-X implantation, an observation that is uncommon in DPO-treated hip dysplasia [4], and, even more so, remained stable at the 12-month evaluation.

Typically, studies on unilateral THR [24,25] and a small cohort of TPO [26] in hip dysplastic dogs demonstrated a significant improvement in the GRFs of operated limbs at 6 months post-operation, a time point not monitored in this study. Force plate analysis in this study revealed a decrease in GRFs after surgery, with levels gradually approaching those observed preoperatively at the 12-month follow-up which is similar to a previous study of three experimental dogs with hip dysplasia using ACE-X surgery [12] reported GRFs reaching baseline levels at 6 months post-surgery. Interestingly, GRFs remained similar to the low baseline values of PVF (40.8 ± 5.9% BW) and P/T index (0.6 ± 0.1) that were close to the values in healthy dogs (PVF 40.4%, P/T index 0.63) [27]. This could be attributed to the use of low-speed walking (0.8–1.3 m/s) during gait analysis, which may not reveal lameness at the preoperative evaluation (visit 1), given that most dogs with HD display lameness after exercise [5,28]. In line with this finding, others also reported the absence of significant differences in PVF and VI between growing dogs with and without HD [29]. In light of the proposed cut-off value of 19.5% asymmetry indexes as a measure of clinically relevant unilateral lameness [30], the present patient cohort, presenting with a 9.6% asymmetry index at the preoperative evaluation, underscores the challenges in detecting substantial changes in gait improvement at early stages of the disease. Despite the absence of detectable improvement at the GRF level and the confounding bias of the owners’ responses due to the open-label design of the study, a decrease in hip joint pain and an improvement in the dogs’ exercise capacity was demonstrated from 1.5 months after treatment, based on the HCPI owner questionnaire. This outcome is in agreement with findings in dogs that underwent THR, wherein 64–82% of owners reported pain relief according to the owner questionnaires [31,32,33]. In contrast, owners reported that the dogs undergoing DPO showed weakness during walking in the early postoperative phase [4].

Based on the radiographic measurements, ACE-X implantation significantly increased the femoral head coverage immediately after surgery, with insignificant changes thereafter. These changes are typically related to variations in positioning and are in line with observations in DPO by Vezzoni et al. [4], who reported NA and PC results showing non-statistically significant decreases at one month post-surgery. Importantly, the measured outcomes consistently fell within the normal range of NA (>105°), LFO (>50%), and PC (>50%) as reported in previous studies [34,35,36]. When femoral head coverage was assessed a year following surgery in relation to different HD treatments, ACE-X surgery showed a greater NA at 132 ± 18° than TPO (116 ± 11°) and DPO (106 ± 4°). Furthermore, the PC after ACE-X surgery (77 ± 20%) was higher than that of DPO (65 ± 6%) but was similar to TPO (78 ± 20%) [37,38]. The ACE-X surgery provided a coverage level at least comparable to or superior to that achieved through TPO and DPO procedures. The ACE-X implant is designed to increase the coverage of the femoral head based on each individual dog’s hip conformation and the design resulted in a dog specific increase in Norberg angle and femoral head coverage. In contrast, the increase in femoral head coverage provided by TPO and DPO relies on the set plate angles available on the market.

In this study, the osteophyte size significantly progressed at the 12-month follow-up compared to the baseline value, increasing by 0.5–1.6 mm, depending on the baseline OA score and the severity of hip laxity. However, a previous study found that osteophyte size increases by 1 mm every three years, as measured by ventrodorsal hip extension radiography [39]. The seemingly faster increase in osteophyte size seen in the current study could be attributed to differences in measurement methods and tools. Here, osteophyte size was measured using 1-millimeter sections of both the coronal and transverse CT of the hip joint, which provided superior details for accurate measurement and improved the early detection of OA [40,41]. When the dogs were categorized based on their baseline OA score in terms of the CT and FCI score on radiographs, those with an OA score of 2 showed the highest progression in osteophyte size (1.6 ± 1.3 mm). Furthermore, the dogs with an FCI score of E exhibited the highest progression in osteophyte size (1.6 ± 1.4 mm). Altogether, these findings suggest that dogs with moderate to severe OA development and/or a high laxity score of the hip joint may not be suitable candidates for ACE-X treatment.

Examination of three implants through X-ray microscopy revealed osseointegration into the porous surface at least by 9 months post-implantation. According to a study on THR, osseointegration of the noncoated 3D-printed porous titanium alloy acetabular cup was observed within a month after implantation [42]. The slower implant–bone integration rates of the noncoated ACE-X implant may be attributed to the direct placement of the implant over the cortical bone of the iliac body. Despite utilizing a periosteal elevator to prompt a periosteal reaction, the mesenchymal stem cells released through this method are probably not as plentiful as those originating from the bone marrow, as observed in the THR procedure exposing the acetabular cancellous bone. Several avenues can improve osteointegration, including (a) osteostimulation (e.g., osteostixis) at the implant placement site to initiate bleeding, facilitating the migration of mesenchymal stem cells to the bone–implant interface [43], (b) the use of surface coatings such as hydroxyapatite or other bioactive materials that mimic the composition of natural bone and promote osteoblast activity, and (c) coatings that modulate the inflammatory response and stimulate bone formation at the implant–bone interface [44,45,46]. It is noteworthy that the available data on osseointegration in the ACE-X implant came from dogs that experienced complications, which could have slowed down the integration of the bone into the implant and produced biased results. Additionally, stability was seen in other ACE-X implants on radiographs and CT scans, indicating that osseointegration may have occurred in clinically sound dogs earlier than the 9-month post-surgery mark.

Complications following ACE-X implantation were reported in this study, with an overall postoperative complication rate of 14.8% and a major complication rate requiring revision of 9.8%. This rate was lower than the previous DPO study in 2010, which reported a 20.7% complication rate [4], but higher than the recent study by the same authors in 2022 which reported an 8.2% complication rate when DPO implants were used instead of TPO implants [47]. Similarly, the reported complication rate for TPO was 33% in an early publication [5] but decreased to 7% in a recent study [48] using locking TPO plates. This decrease can be attributed to surgeons gaining more experience and advancements in surgical techniques and implants. Although the complication rate of ACE-X is higher than in the most recent studies of both DPO and TPO, it is lower than in the initial reports for both techniques. The lower incidence of complications in the ACE-X study compared to the initial pelvic osteotomy studies may be attributed to the less invasive surgical approach, which eliminates the need for osteotomy to reshape the acetabulum, preservation of the natural pelvic canal, and absence of the risk of delayed or non-union of bone. Risk factors associated with severe complications following ACE-X implantation included pronounced hip laxity, accelerated OA progression, and marked forelimb lameness. Moreover, in the event of ACE-X failure, a salvage THR presents a viable alternative that yields positive outcomes, as opposed to DPO/TPO, where implant failure may lead to severe complications, the irreversible impairment of limb functions, and the added challenge of revision surgery [8]. Of note, three implant failures were observed in this study. Two failures occurred in a dog with bilateral humeral condyle fractures five months post-ACE-X implantation. The third failure was in a dog with contralateral hip luxation, who underwent THR six weeks after ACE-X implantation. In these patients, implant failure may result from the overstress placed on the implant as a result of implant overload. Therefore, ensuring critical case selection at the time of inclusion and preventing concurrent injuries during the initial osseointegration phase is essential for the long-term success of the implant.

This study has several limitations to consider. This being the initial clinical investigation employing the ACE-X implant in client-owned dogs, a wide spectrum of dogs affected by HD were enrolled, which might have contributed to a higher complication rate due to inappropriate case selection (i.e., moderate OA development and severe hip laxity). The advancement of OA to a severe stage in certain dogs was also influenced by a delay in implant manufacturing, possibly exacerbated by COVID-related disruptions. Furthermore, 26% of dog patients did not complete the 12-month follow-up, which could have introduced a bias to the results. Hence, future studies should implement strict inclusion criteria, concentrating on dogs with mild to moderate hip laxity and minimal or no preoperative OA development, as determined by preoperative CT scans. Efforts should be made to reduce the manufacturing (lead) time of the implant to 2–4 weeks. Additionally, a combination of owner questionnaires or client-specific outcome measures, along with performing force plate analysis after exercise, could be considered to better clarify and quantify the reduction in pain-related activities. Moreover, given the intended lifelong placement of ACE-X implants in dogs, extended follow-ups utilizing owner questionnaires and imaging surveillance are imperative for accurately assessing long-term outcomes.

## 5. Conclusions

In conclusion, the utilization of the ACE-X implant in dogs with hip dysplasia offers a comprehensive array of benefits. These include increased coverage of the femoral head, improved support and stability for the dysplastic hip joint, alleviated hip pain and dysfunction, reduced surgical and recovery durations, and reduced postoperative complications due to a less invasive surgical approach. The enhanced joint stability provided by the ACE-X implant diminishes the risk of subluxation or dislocation, ultimately leading to decreased discomfort and dysfunction in HD-affected dogs. The customization and precision of ACE-X implants, tailored to individual canine anatomy, provide an optimal fit, thereby minimizing complications and maximizing effectiveness. Furthermore, the reduced invasiveness of ACE-X implantation, preserving the joint and minimizing soft tissue trauma, allows for simultaneous bilateral procedures and necessitates a shorter postoperative recovery period compared to traditional methods. Notably, the lower reported complication rate associated with ACE-X implantation, which may be further minimized by strict inclusion criteria and the reduction of implant lead time, renders it a valuable and promising treatment option for enhancing joint stability, correcting dysplastic anatomy, and improving overall quality of life in young dogs with hip dysplasia. 

## Figures and Tables

**Figure 1 animals-14-02385-f001:**
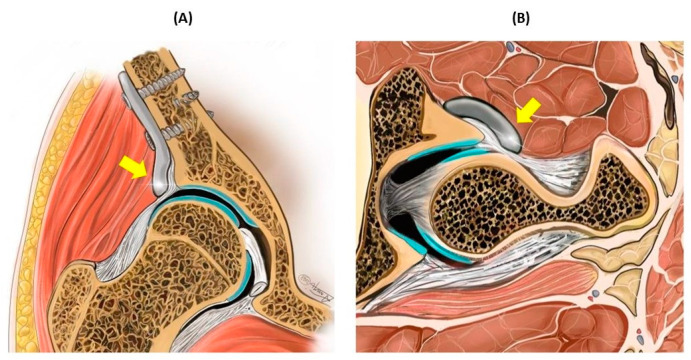
Schematic representation of the dysplastic hip joint with the acetabular rim extension (ACE-X) implant in the coronal plane (**A**) and transverse plane (**B**). The ACE-X implant (indicated by the yellow arrow) was positioned extracapsular of the hip joint and extended to the dorsal acetabular rim, thereby increasing femoral head coverage.

**Figure 2 animals-14-02385-f002:**
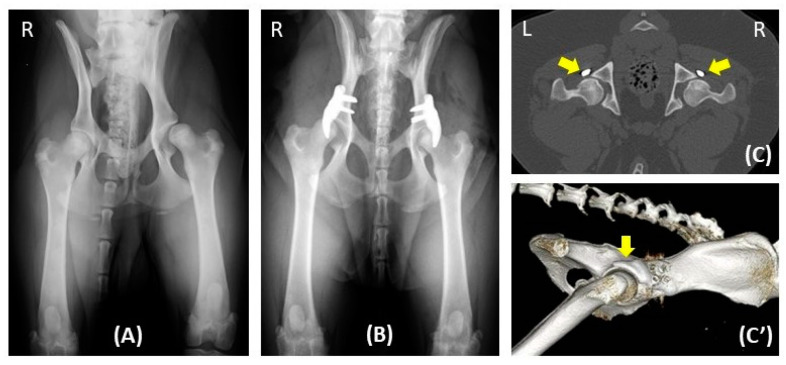
One-year radiographic follow-up in a dog with hip dysplasia undergoing bilateral acetabular rim extension (ACE-X) surgery. Preoperative radiography (**A**) reveals hip laxity without evidence of osteoarthritis (OA). Immediate postoperative radiography (**B**) after bilateral ACE-X implantation shows the minimal development of OA (Morgan line) during the lead time. Transverse CT of the hips (**C**) and 3D reconstruction of the right lateral pelvis (**C’**) at the 1-year follow-up highlight the rim-extension part of the ACE-X implants (yellow arrow), extending the dorsal acetabular rim and enhancing femoral head coverage without the progression of OA.

**Figure 3 animals-14-02385-f003:**
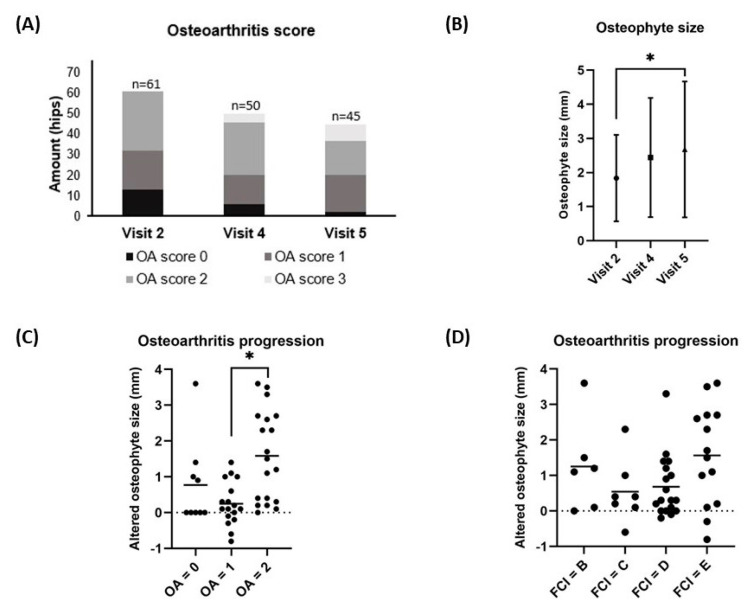
The changes in osteoarthritis (OA) score (**A**) and osteophyte size (**B**) from the baseline value (visit 2) to the 12-month follow-up (visit 5), and the comparison of osteophyte size growth between visit 2 and visit 5, grouped by baseline OA score (**C**) and FCI score (**D**), are depicted. (**A**) Each bar represents the number of hips in each OA score at each time point. (**B**) Each dot represents the mean osteophyte size ± SD at each time point. (**C**,**D**) Each dot represents an individual change in osteophyte size over 12 months, while each line illustrates the mean osteophyte size change over 12 months. * A *p*-value of <0.05 was considered significant using Bonferroni correction for multiple tests. OA score progression was based on CT measurements of the osteophytes from baseline (visit 2) to visit 5. The FCI score was designated from pre-operative radiographs at visit 1.

**Figure 4 animals-14-02385-f004:**
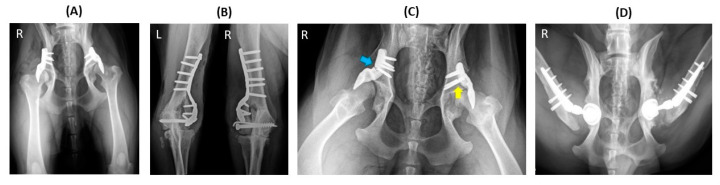
Implant failure of hip no. 23 (left hip) and hip no. 34 (right hip) in a dog that underwent staged bilateral dorsal acetabular rim extension (ACE-X) with a two-month interval. (**A**) The immediate postoperative radiograph showed no complication of the implants’ placement, with moderate hip osteoarthritis (OA). Five months after the second ACE-X, the dog experienced bilateral humeral condyle fractures requiring humeral plating and screw fixation (**B**), which was followed by a broken ACE-X implant (blue arrow), broken screws (yellow arrow) (**C**), and severe OA progression, requiring revision to staged bilateral total hip replacement (**D**).

**Figure 5 animals-14-02385-f005:**
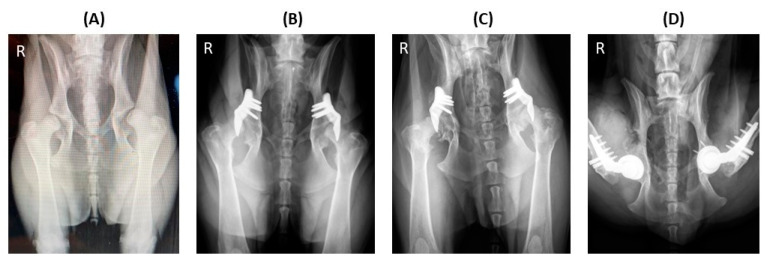
Severe progression of osteoarthritis (OA) after bilateral dorsal acetabular rim extension (ACE-X; hip nos. 24 and 25). The preoperative radiograph (**A**) shows hip dysplasia with hip laxity without OA. Following a 49-day ACE-X lead time, the immediate postoperative radiograph (**B**) shows an increase in laxity and osteophyte formation at both femoral necks. The 10-month follow-up radiograph (**C**) indicates the severe progression of OA, which required staged bilateral total hip replacement (**D**).

**Figure 6 animals-14-02385-f006:**
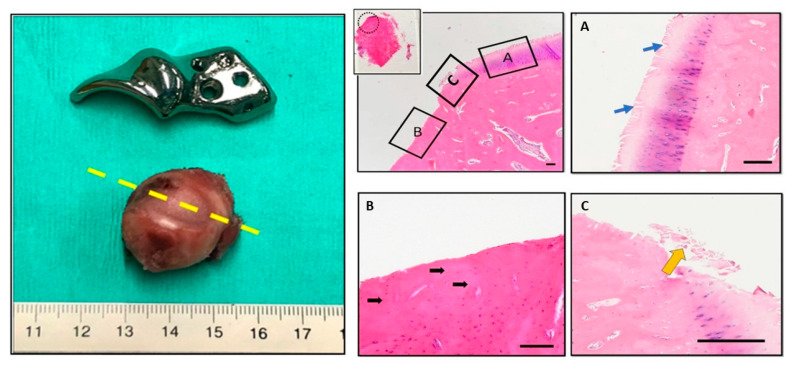
Gross examination and histopathological findings of the right femoral head (hip no. 37). The excised femoral head shows significant cartilage loss, corresponding to the rim extension part of the ACE-X implant. Histology sections made at the level of cartilage loss (yellow dash line) and stained with H&E display erosions and denudation of the femoral head. Magnified fields (dotted-line circle and rectangular boxes) reveal specific features: Box (**A**) highlights areas of articular cartilage, blue arrows depict the extension of matrix fibrillation downward and the presence of vertical fissures, Box (**B**) indicates denudation and complete erosion of the unmineralized hyaline cartilage; black arrows point to empty lacunae, suggesting the loss of osteocytes within the subchondral bone tissue, and Box (**C**) indicates the debris of necrotic cartilage tissue (yellow arrow).

**Figure 7 animals-14-02385-f007:**
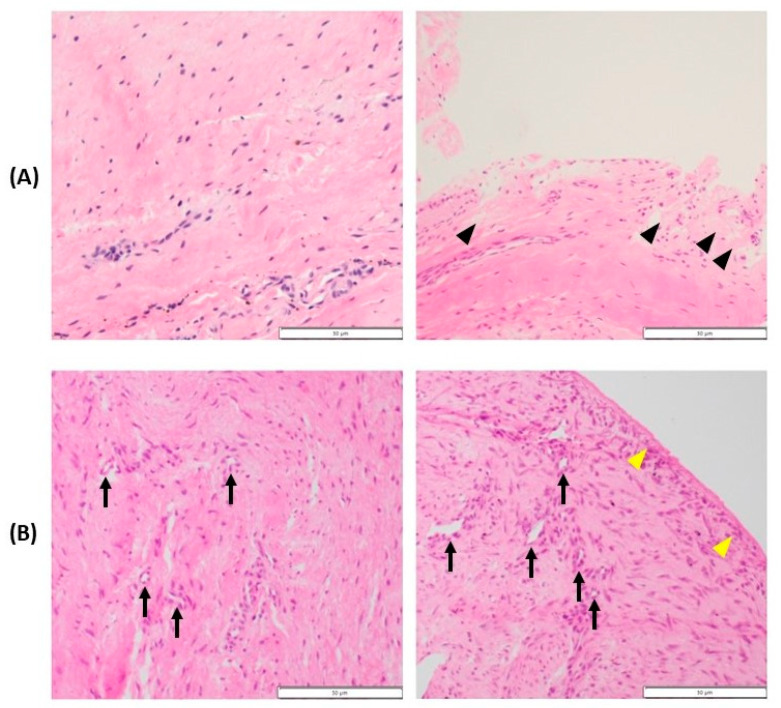
Histopathology of the synovium of hip no. 34 (**A**) and hip no. 37 (**B**). (**A**) Representative images showing slight activation of the synovial stroma and significant subintimal edema (black arrowheads). (**B**) Micrographs reveal hyperplasia of the synovial intima (yellow arrowheads), marked activation of the synovial stroma (densely located fibroblast and histocytes), and increased vascularization (black arrows).

**Figure 8 animals-14-02385-f008:**
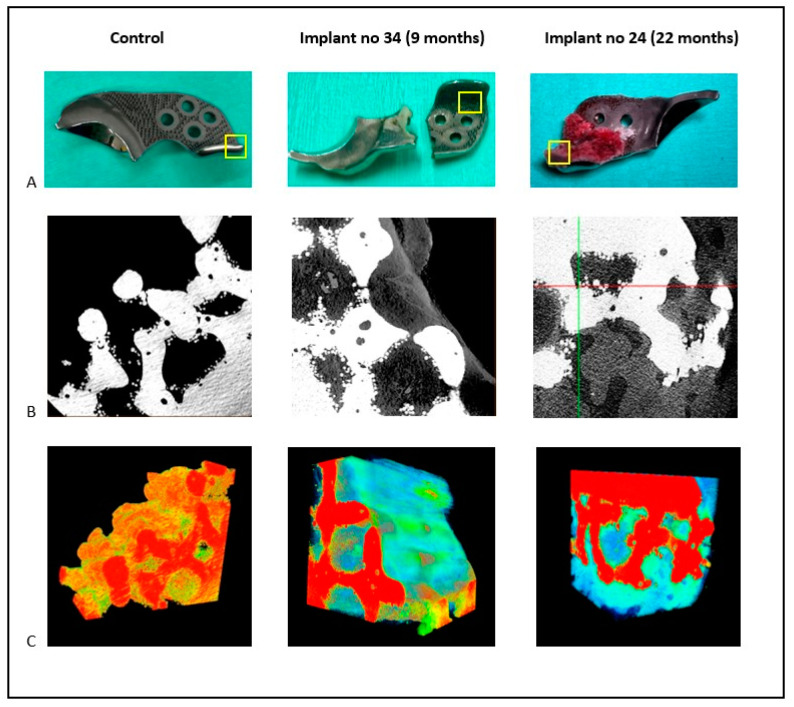
The 3D X-ray microscopy images of both the control and extracted ACE-X implants (hip nos. 34 and 24). (**A**) Images of the implants highlight the study area within the yellow box. (**B**) Original 2D imaging displaying varying density areas in gray values. White indicates the high-density zone, while black indicates the low-density zone. (**C**) A 3D reconstruction of the studied area, with color visualization representing different density areas. Red indicates the highest density zone, followed by orange, yellow, green, and blue. Trabeculae are depicted in light or dark blue, depending on the bone tissue density.

**Figure 9 animals-14-02385-f009:**
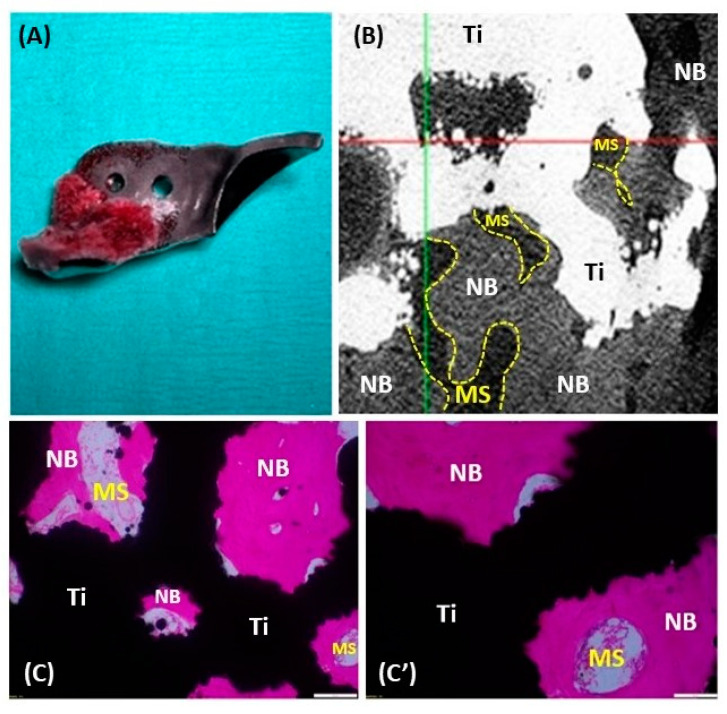
Histology of the removed ACE-X implant (hip no. 24) showing osseointegration at the porous surface. (**A**) The removed ACE-X implant from hip no. 24 at 22 months after surgery. (**B**) A 2D X-ray microscopy image demonstrating new bone formation inside the porous structure of the implant. (**C**,**C’**) Histological images highlighting the newly formed bone (NB), bone marrow space (MS), and titanium implant (Ti).

**Table 1 animals-14-02385-t001:** Study outline.

Evaluation	Preoperative Day (Visit 1)	Day of Surgery (Visit 2)	1.5-Month Follow-Up (Visit 3)	3-Month Follow-Up (Visit 4)	12-Month Follow-Up (Visit 5)
Radiograph	x ^a^	x	x	-	-
CT scan	x ^b^	x ^b^	-	x ^b^	x ^b^
Osteophyte size *	x	x	-	x	x
OA scoring **	x ^a,b^	x	-	x	x
FCI scoring	x	-	-	-	-
Force plate gait analysis	x	-	x	x	x
HCPI	x	-	x	x	x
Ortolani’s test	x	x	x	x	x

CT: computed tomography, OA: osteoarthritis, FCI: Fédération Cynologique Internationale, HCPI: Helsinki chronic pain index. * Osteophyte size was measured on CT scans at three locations: the femoral neck and the cranial and caudal acetabular rim. ** At intake, OA was scored using a preoperative radiograph ^a^ for inclusion. For monitoring, the OA score was based on the maximum osteophyte size, determined through CT measurements ^b^.

**Table 2 animals-14-02385-t002:** Clinical tests were conducted per visit. The numbers are displayed as performed tests/total tests (percentage).

Evaluation	Visit 1	Visit 2	Visit 3 (24–92 Days)	Visit 4 (68–289 Days)	Visit 5 (270–553 Days)
CT scan *	61/61 (100%)	61/61 (100%)	-	50/61 (82%)	45/61 (74%)
OA scoring *	61/61 (100%)	61/61 (100%)	-	50/61 (82%)	45/61 (74%)
Force plate *	51/61 (84%)	-	54/61 (89%)	46/61 (75%)	44/61 (72%)
HCPI **	33/41 (80%)	-	21/41 (51%)	23/41 (56%)	25/37 ^Ɨ^ (68%)
Ortolani’s test *	61/61 (100%)	61/61 (100%)	55/61 (90%)	50/61 (82%)	45/61 (74%)

HCPI: Helsinki chronic pain index. * The quantity represents the number of hips. ** The quantity represents the number of dogs. ^Ɨ^ The 4 dogs that underwent surgery on bilateral hips, with a 3-month interval in between, returned for a 1-year follow-up to assess both hips in a single visit, bringing the total number to 37 dogs.

**Table 3 animals-14-02385-t003:** Results from dogs with hip dysplasia that underwent acetabular rim extension and were assessed preoperatively (visit 1), immediately postoperatively (visit 2), at 1.5 months (visit 3), at 3 months (visit 4), and at 12 months (visit 5) of follow-up.

Outcome Measurements		Visit 1	Visit 2	Visit 3	Visit 4	Visit 5	*p*-Value
CT scan measurements (Mean ± SD)	NA (°)	88.5 ± 12.6 ^a^	137.6 ± 19.2 ^b^	-	134.3 ± 19.0 ^b^	131.5 ± 17.8 ^b^	<0.000
	LFO (%)	24.3 ± 16.2 ^a^	83.8 ± 16.4 ^b^	-	79.8 ± 18.2 ^b^	78.1 ± 17.4 ^b^	<0.001
	PC (%)	34.7 ± 17.3 ^a^	82.3 ± 20.0 ^b^	-	80.2 ± 18.8 ^b^	77.4 ± 19.5 ^b^	<0.001
GRFs (Mean ± SD)	PVF (%BW)	40.9 ± 5.8	-	38.9 ± 4.8	39.8 ± 5.4	40.9 ± 6.0	0.166
	P/T index	0.62 ± 0.1	-	0.58 ± 0.1	0.59 ± 0.1	0.61 ± 0.1	0.067
	VI (%BW.s)	14.2 ± 3.0	-	13.8 ± 3.4	14.3 ± 3.0	14.6 ± 4.0	0.705
	Breaking force (%BW)	6.38 ± 1.7	-	6.11 ± 1.6	6.41 ± 1.5	6.43 ± 1.9	0.726
	Propulsion force (%BW)	6.64 ± 1.5 ^a^	-	5.99 ± 1.6 ^a,b^	5.78 ± 1.4 ^b^	5.58 ± 1.9 ^b^	0.007
HCPI (%) (Mean ± SD)		30.29 ± 13.55 ^a^	-	21.86 ± 12.39 ^b^	19.37 ± 11.54 ^b^	18.27 ± 12.0 ^b^	0.002

NA: Norberg angle, LFO: linear percentage of femoral head overlap, PC: percentage of femoral head coverage, SD: standard deviation, GRFs: ground reaction forces, PVF: peak vertical force, P/T index: pelvic/thoracic index, VI: vertical impulse, BW: body weight, s: second, ^a,b^
*p*-value < 0.05 based on post hoc Bonferroni correction, *p*-value < 0.05 from the generalized linear mixed model. HCPI (%) = 100% x (total index score/maximum possible index score of the answered questions).

**Table 4 animals-14-02385-t004:** Comprehensive details of the four dogs (six hips) exhibiting major complications resulting in lameness after dorsal acetabular rim extension.

Parameters		Dog 1		Dog 2	Dog 3		Dog 4
Breed		Newfoundland		Border Collie	German Shepherd dog		Bernese mountain dog
Age at surgery (m)		9	12	7	8		12
Body weight (kg)		38.6	43.7	13.8	26.8		35.6
Hip no.		23	34	37	24	25	53
Side		left	right	right	right	left	right
Lead time (days)		122	107	48	49		74
FCI score		D	E	E	D	E	B
Preoperative NA (°)		87	83	82	94	86	104
Osteophyte size (mm) *	Visit 1	2.4	3.4	1.7	1.8	1.9	0
	Visit 2	4.2	4.4	2.3	4	3	0
	Last CT (days) **	6.3 (68)	n/a	n/a	7.3 (336)	5.3(91)	3.6 (528)
OA scoring *	Visit 1	2	2	1	1	1	0
	Visit 2	2	2	2	2	2	0
	Last CT	3	n/a	n/a	3	3	2
Complication		2 broken screws and severe OA	broken implant and severe OA	4 broken screws	severe OA	severe OA	septic arthritis
Follow-up time until complication (days)		246	178	215	291	291	528

m: months, kg: kilograms, FCI: the Fédération Cynologique Internationale, NA: Norberg angle, mm: millimeters, OA: osteoarthritis, and n/a: not available. * Osteophyte size and OA score were assessed using CT. ** The osteophyte size, measured from the last CT (the time interval between the surgery day and when the final CT scan was completed).

## Data Availability

The data presented in this study are available within the article. Raw data supporting this study are available from the corresponding author by request.

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
