# Peer review of "Outcome One Year after Acetabular Rim Extension Using a Customized Titanium Implant for Treating Hip Dysplasia in Dogs"

_animals, 2024, doi:10.3390/ani14162385_

Round 1

Reviewer 1 Report

Comments and Suggestions for Authors

This manuscript is very interesting. The authors did a great job, and the results are encouraging. I want to highlight one thing only. According to the literature, DPO results are not very bad. 

line 565 - It is written that the complication rate after the DPO procedure is 20,7%, based on the: 4. Vezzoni A, Boiocchi S, Vezzoni L, Vanelli AB, Bronzo V. Double pelvic osteotomy for the treatment of hip dys- 659 plasia in young dogs. Vet Comp Orthop Traumatol. (2010) 23(6):444-52. doi: 10.3415/VCOT-10-03-0034

But searching the literature you cannot omit another result, showing that DPO technique is not very hazardous - just an example - with  0,6% of major complications and 6% of minor ones:  Vet Comp Orthop Traumatol 2022; 35(01): 047-056 DOI: 10.1055/s-0041-1735288. 

Similar comment to the TPO complications rate. Authors indicate 33%, but the cited paper is very old. This should be verified according to the recent knowledge. 

Line 71 – I would be more cautious in assessing whether the method gives more predictable results. One year of clinical observation is too short to state this. After DPO/TPO treatment  signs of OA are seen usually after 4-5-6 years.

Line 116 and line 241-  dogs older than 6 month. The question is – why Authors did not choose dogs in the same age. That will be more accurate to compare results. It is difficult to compare 6 month -old dog with 38 month-old dog, because the first one is still growing, the second one has finished growing time a long time ago.

Line 248 – the median lead time between the preoperative day and surgery day is the weak point of this protocol, and must have had an impact on the condition of the hip joint. In dogs with hip dysplasia, with laxity in joints the average waiting time – 74 days – certainly influenced the morphology of the dorsal acetabular rim and the shape of the acetabulum. The prepared implant would not fit the bone measured 2 months earlier. The longest lead time was 158 days – I expected that in such situation the joint must have changed and planned implant may have not been precise enough for this particular dog.

Line 252 – would be more informative and clear to have a graph/table  showing number of the patient, age, lead time

Line 531 – it is said that ACE-X surgery provides a coverage level at least comparable to or superior to that after TPO or DPO procedure. The question is – does it depend on the plate angle? Or what kind of plates these results were compared to?

Line 569 – complication rate after TPO/DPO depends on the citation. For example in 2022 there is a publication in which authors had a group of  458 dogs and in this group they faced with 8,2% of complications.
So, this part should be rewrite, to be sure that readers will have a wide spectrum of real postsurgical results after DPO/TPO

Author Response

Please see the rebuttal letter in the attachment.

Comments 1: line 565 - It is written that the complication rate after the DPO procedure is 20,7%, based on the: 4. Vezzoni A, Boiocchi S, Vezzoni L, Vanelli AB, Bronzo V. Double pelvic osteotomy for the treatment of hip dys- 659 plasia in young dogs. Vet Comp Orthop Traumatol. (2010) 23(6):444-52. doi: 10.3415/VCOT-10-03-0034

But searching the literature you cannot omit another result, showing that DPO technique is not very hazardous - just an example - with  0,6% of major complications and 6% of minor ones:  Vet Comp Orthop Traumatol 2022; 35(01): 047-056 DOI: 10.1055/s-0041-1735288. 

Similar comment to the TPO complications rate. Authors indicate 33%, but the cited paper is very old. This should be verified according to the recent knowledge.

Response 1: Thank you for pointing this out. We agree with this comment. However, considering that this study is the first report using this surgical technique, the complication rate we found was lower than the initial reports for both TPO and DPO. Therefore, we have revised the main text between lines 570–581 to “This rate was lower than the previous DPO study in 2010, which reported a 20.7% complication rate [4], but higher than the recent study by the same authors in 2022 which reported an 8.2% complication rate when DPO implants were used instead of TPO implants [47]. Similarly, the reported complication rate for TPO was 33% in an early publication [5], but decreased to 7% in a recent study [48] using locking TPO plates. This decrease can be attributed to surgeons gaining more experience and advancements in surgical techniques and implants. Although the complication rate of ACE-X is higher than in the most recent studies of both DPO and TPO, it is lower than in the initial reports for both techniques. The lower incidence of complications in the ACE-X study compared to the initial pelvic osteotomy studies may be attributed to the less invasive surgical approach, which eliminates the need for osteotomy to reshape the acetabulum, preservation of the natural pelvic canal, and absence of the risk of delayed or non-union of bone.”

We expect that the complication rate with the ACE-X technique will further decrease in the future as we implement stricter patient inclusion criteria and gain more experience with this surgical method, similar to what has been observed with DPO and TPO.

    We added the new references related to the recent report for DPO and TPO on page 20 between Line 782 – 785 as follows:

  “47. Tavola F, Drudi D, Vezzoni L, Vezzoni A. Postoperative Complications of Double Pelvic Osteotomy Using Specific Plates in 305 Dogs. Vet Comp Orthop Traumatol. (2022) 35(1):47-56. doi: 10.1055/s-0041-173528

  1. Rose SA, Bruecker KA, Petersen SW, Uddin N. Use of locking plate and screws for triple pelvic osteotomy. Vet Surg. (2012) 41(1):114-20. doi: 10.1111/j.1532-950X.2011.00927.x"

Comments 2: Line 71 – I would be more cautious in assessing whether the method gives more predictable results. One year of clinical observation is too short to state this. After DPO/TPO treatment  signs of OA are seen usually after 4-5-6 years.

Response 2: Thank you for pointing this out. We agree with this comment. Therefore, we have removed “more predictable outcome” at Line 71 and changed it to “with immediate reduction of hip laxity”.

Comment 3: Line 116 and line 241-  dogs older than 6 month. The question is – why Authors did not choose dogs in the same age. That will be more accurate to compare results. It is difficult to compare 6 month -old dog with 38 month-old dog, because the first one is still growing, the second one has finished growing time a long time ago.

Response 3: Since this is the first study of this surgical technique in clinical patients, we aimed to include a wide spectrum of patients who still tested positive for Ortolani's sign with absent or minimal evidence of hip osteoarthritis. Therefore, a maximum age limit was not set but a minimum age limit was set due to the presence of an open acetabular growth plate, which may affect the implant design and lead to inaccurate fitting of the implant.

Comment 4: Line 248 – the median lead time between the preoperative day and surgery day is the weak point of this protocol, and must have had an impact on the condition of the hip joint. In dogs with hip dysplasia, with laxity in joints the average waiting time – 74 days – certainly influenced the morphology of the dorsal acetabular rim and the shape of the acetabulum. The prepared implant would not fit the bone measured 2 months earlier. The longest lead time was 158 days – I expected that in such situation the joint must have changed and planned implant may have not been precise enough for this particular dog.

Response 4: We agree that the long lead time after the design affected the accurate fitting of the implant to the bone due to the progression of osteoarthritis and joint capsule fibrosis, as we reported in a previous study, where the osteophyte size significantly increased during the lead time (Kwananocha et al., 2023). The extended lead time in this study was due to several factors, including delays in implant production and clinical workflow caused by the COVID-19 pandemic, the owners' personal decisions to postpone the surgery, and the staged bilateral hip surgeries in the first few dogs. Currently, the lead time has been shortened to 2-4 weeks and inclusion criteria have been adapted, e.g. minimal OA requirement is determined on CT and not on radiograph. Regarding implant fitting, the accuracy of implant placement is under investigation and findings will be a topic for a future manuscript.

Comment 5: Line 252 – would be more informative and clear to have a graph/table showing number of the patient, age, lead time

Response 5: Thank you for your recommendation. We appreciate the suggestion to create a more informative demographic findings table. However, given that our manuscript already included 13 figures and tables, we decided to describe patient demographics and lead time more efficiently in the manuscript text.

Comment 6: Line 531 – it is said that ACE-X surgery provides a coverage level at least comparable to or superior to that after TPO or DPO procedure. The question is – does it depend on the plate angle? Or what kind of plates these results were compared to?

Response 6: Yes, the post-operative Norberg angle (NA) and percentage of femoral head coverage (PC) in TPO and DPO techniques depend on the plate angle, which is available in specific sizes (20°, 25°, 30°, and 35°) on the market. In contrast, the ACE-X implant can increase the coverage of the femoral head to varying degrees based on the individual dog's needs.

In the DPO study by Petazzoni (2022), a 20° DPO plate was used in 8 hips, a 25° DPO plate in 1 hip, a 20° TPO plate in 1 hip, and a 25° TPO plate in 1 hip. The plate selection was based on the preoperative DAR view. The study found an increase in the NA from 87° to 106° and an increase in the PC from 24% to 65% one year after surgery. When comparing the differences in NA before and after surgery, the NA increased by 19°, which correlated with the most frequently used implant angle.

In the TPO study by Borostyankoi (2003), adjustable iliac bone plates of 25° and 35° were used based on the preoperative reduction angle. The specific number of each plate angle used was not reported. The study found an increase in the NA from 93° to 116° and an increase in the PC from 43% to 78% one year after surgery. When comparing the differences in NA before and after surgery, the NA increased by 23°. Unfortunately, the study did not specify which implant angle was used predominantly.

To clarify this statement, we have revised the manuscript between lines 528-533 on page 15 to “The ACE-X surgery provided a coverage level at least comparable to or superior to that achieved through TPO and DPO procedures. The ACE-X implant is designed to increase the coverage of the femoral head based on each individual dog's hip conformation and the design resulted in a dog specific increase in Norberg angle and femoral head coverage. In contrast, the increase in femoral head coverage provided by TPO and DPO relies on the set plate angles available on the market.”

Comment 7: Line 569 – complication rate after TPO/DPO depends on the citation. For example in 2022 there is a publication in which authors had a group of  458 dogs and in this group they faced with 8,2% of complications.
So, this part should be rewrite, to be sure that readers will have a wide spectrum of real postsurgical results after DPO/TPO

Response 7: Please see also the answer to comment 1.

Thank you for pointing this out. We agree with this comment. However, considering that this study is the first report using this surgical technique, the complication rate we found was lower than the initial reports for both TPO and DPO. Therefore, we have revised the main text between lines 570–581 to “This rate was lower than the previous DPO study in 2010, which reported a 20.7% complication rate [4], but higher than the recent study by the same authors in 2022 which reported an 8.2% complication rate when DPO implants were used instead of TPO implants [47]. Similarly, the reported complication rate for TPO was 33% in an early publication1[5], but decreased to 7% in a recent study [48] using locking TPO plates. This decrease can be attributed to surgeons gaining more experience and advancements in surgical techniques and implants. Although the complication rate of ACE-X is higher than the most recent studies of both DPO and TPO, it is lower than in the initial reports for both techniques. The lower incidence of complications in the ACE-X study compared to the initial pelvic osteotomy studies may be attributed to the less invasive surgical approach, which eliminates the need for osteotomy to reshape the acetabulum, preservation of the natural pelvic canal, and absence of the risk of delayed or non-union of bone.”

We expect that the complication rate with the ACE-X technique will further decrease in the future as we implement stricter patient inclusion criteria and gain more experience with this surgical method, similar to what has been observed with DPO and TPO.

We added the new references related to the recent report for DPO and TPO on page 20 between Line 782 – 785 as follows:

  “47. Tavola F, Drudi D, Vezzoni L, Vezzoni A. Postoperative Complications of Double Pelvic Osteotomy Using Specific Plates in 305 Dogs. Vet Comp Orthop Traumatol. (2022) 35(1):47-56. doi: 10.1055/s-0041-173528

  1. Rose SA, Bruecker KA, Petersen SW, Uddin N. Use of locking plate and screws for triple pelvic osteotomy. Vet Surg. (2012) 41(1):114-20. doi: 10.1111/j.1532-950X.2011.00927.x”

Reviewer 2 Report

Comments and Suggestions for Authors

This pioneering study evaluated the clinical outcomes of dogs over a 1-year follow-up period after receiving a personalized 3D-printed titanium implant for dorsal Acetabular Rim Extension (ACE-X) to treat hip dysplasia. The study is well-written, with various information presented in tables or illustrated in figures. I only have a few comments.  

Line 147: Why was the baseline value for the OA score and the measured osteophyte size established from the immediate postoperative CT scan at visit 2 and not from visit 1?

Line 174: what kind of further analysis? You changed the scoring system to percentage. I believe this could potentially introduce a confounding factor.

Line 179: It would be interesting to know the type of sedation utilized; I presume that dogs were submitted to general anesthesia for the Ortolani’s test only because it was necessary for other procedures such as CT scans, is it right? Perhaps this clarification would be valuable.

Line 479: Based on the evaluations conducted in this study (primarily the Helsinque questionnaire), is it appropriate to conclude that the implant reduced pain-related exercise limitations?

Line 496: returned?

Line 582: Limitations - Would you consider modifying or improving the types of evaluations conducted? For example, those related to exercise tolerance/intolerance,

Lines 612-613 – I think it would be overly optimistic to conclude that the implant is safe and valuable before a long-term evaluation, considering that this type of implant is only recommended for young animals. Therefore, it is essential to assess the long-term results before drawing these conclusions…

Author Response

Please see the rebuttal letter in the attachment.

Comments 1: Line 147: Why was the baseline value for the OA score and the measured osteophyte size established from the immediate postoperative CT scan at visit 2 and not from visit 1?

Response 1: Thank you for pointing this out. Based on our previous study (Kwananocha, et al. 2023), we observed a significant increase in osteophyte size during the lead time (46-158 days). To prevent the confounding factor of the variable lead time on OA progression, we have redesigned our study to use computed tomography (CT) imaging taken immediately post-operatively (visit 2) as the baseline value for osteophyte measurement in this study. This adjustment aims to minimize the impact of OA progression during the lead time on our assessment of the ACE-X implant's effectiveness in slowing or stopping the progression of OA.

Comments 2: Line 174: what kind of further analysis? You changed the scoring system to percentage. I believe this could potentially introduce a confounding factor.

Response 2: Thank you for pointing this out. We apologize for not initially providing a specific reason for using the percentage score in this study. The Helsinki Chronic Pain Index (HCPI) Questionnaire consists of 11 queries. However, one of these queries pertains to the ability to perform high-speed running, which is not recommended during the six weeks post-surgery. Consequently, some owners opted to leave this query unanswered, making it impractical to use the total sum score for analysis. To ensure comparability of the HCPI results at the six-week control with those from other visits, we normalized the scores to a percentage of the answered questions, following this formula:

HCPI (%) = 100% x (total index score/maximum possible index score of the answered questions).

We then also calculated the reference value to percentage for interpretation of the results. With this normalization, we believe we decreased the confounding factor of incomplete questionnaires for data analysis instead of using the routine method (sum score).

Comments 3: Line 179: It would be interesting to know the type of sedation utilized; I presume that dogs were submitted to general anesthesia for the Ortolani’s test only because it was necessary for other procedures such as CT scans, is it right? Perhaps this clarification would be valuable.

Response 3: Sorry for the confusion. The primary purpose of sedation and general anesthesia was not to perform the Ortolani's test (this was also done during orthopedic exam in many of the awake dogs), however, we ensured the Ortolani’s test was conducted and repeated during the procedures requiring sedation and anesthesia. Specifically, general anesthesia was used for CT scans at visits 1, 4, and 5, and for surgery at visit 2, while sedation was used for x-rays at visit 3. Unfortunately, the protocols or medications used for sedation and anesthesia varied based on the dog's ASA score, the anesthesiologists' preferences, and the specific requirements of the procedures (CT, x-ray, or surgery). To emphasize that sedation and anesthesia were not used solely for Ortolani’s test, the manuscript was revised between Lines 179-181 on page 5 as follows: “The Ortolani's test was performed while the dog was in lateral recumbency, under either sedation or general anesthesia, as required for the  CT scans, radiographs, and surgery during each of the five scheduled visits.”

Comments 4: Line 479: Based on the evaluations conducted in this study (primarily the Helsinque questionnaire), is it appropriate to conclude that the implant reduced pain-related exercise limitations?

Response 4: The HCPI questionnaire consists of 11 queries related to mood, willingness to play, pain expression, and abilities to move and exercise. We observed a significant decrease in the HCPI score, from 30% pre-operatively to 18% post-operatively, indicating a reduction in pain expression and an improvement in the dogs’ movement and exercise capabilities. According to Hielm-Björkman's publication in 2003 and 2009, a cumulative HCPI score over 25% (score of 11) signifies chronic pain. In this study, the pre-operative score of 30% indicated chronic pain, but it gradually decreased to 18%. Although this was not lower than the threshold of 13.6% to indicate pain-free, most owners were satisfied with the outcome and reported that their dogs could perform all activities without limitations or better than before surgery, except in cases with complications. Furthermore, some owners noted that their dogs could engage in activities such as jumping and standing on their hind legs post-surgery, which they had never done before surgery. Based on this feedback from owners and the improvement in HCPI scores, we are confident that the implant increases the stability of the hip joint, thereby reducing pain during exercise in hip dysplastic dogs. 

Comments 5: Line 496: returned?

Response 5: The sentence was rewritten and we hope this addresses the question. We observed a non-significant decrease in the majority of the force plate data at six weeks post-surgery. However, these values gradually increased over time and eventually reached similar levels as those observed on the preoperative day. We have revised the manuscript between Lines 496-500 to “Force plate analysis in this study revealed a decrease in GRFs after surgery, with levels gradually approaching those observed pre-operatively at the 12-month follow-up which is similar to a previous study of three experimental dogs with hip dysplasia using ACE-X surgery [12] reported GRFs reaching baseline levels at 6 months post-surgery.”

Comments 6: Line 582: Limitations - Would you consider modifying or improving the types of evaluations conducted? For example, those related to exercise tolerance/intolerance

Response 6: Thank you for pointing this out. For a future study, we will consider using a combination of owner questionnaires to better clarify and quantify the reduction in pain-related activities. For this we have added the LOAD questionnaire to the future protocol. Additionally, we may use client-specific outcome measures to monitor specific impairments in each dog. Furthermore, we plan to incorporate force plate analysis as an objective assessment to measure ground reaction forces after exercise at a trotting speed, which increases the possibility of detecting lameness or improvements in ground reaction forces. We added the recommended future assessment into the manuscript between Lines 606-608 as follows: “Additionally, a combination of owner questionnaires or client-specific outcome measures, along with performing force plate analysis after exercise, could be considered to better clarify and quantify the reduction in pain-related activities.”

Comments 7: Lines 612-613 – I think it would be overly optimistic to conclude that the implant is safe and valuable before a long-term evaluation, considering that this type of implant is only recommended for young animals. Therefore, it is essential to assess the long-term results before drawing these conclusions.

Response 7: The authors agree with this comment and removed “Consequently, the ACE-X implant emerges as a secure and valuable option for clinical application.” between Lines 628-629 on page 17 and revised the conclusion in the abstract section between Lines 45-48 to “In conclusion, the ACE-X implant effectively increased femoral head coverage, eliminated subluxation, and provided pain relief with minimal complications, benefiting over 90% of the study population. The study supports the ACE-X implant as a valuable alternative treatment for canine hip dysplasia.”
